# Total Content of Saponins, Phenols and Flavonoids and Antioxidant and Antimicrobial Activity of In Vitro Culture of *Allochrusa gypsophiloides* (Regel) Schischk Compared to Wild Plants

**DOI:** 10.3390/plants12203521

**Published:** 2023-10-10

**Authors:** Valentina K. Mursaliyeva, Balaussa T. Sarsenbek, Gulnara T. Dzhakibaeva, Tlek M. Mukhanov, Ramazan Mammadov

**Affiliations:** 1Institute of Plant Biology and Biotechnology, Almaty 050040, Kazakhstan; s.balausa.s@mail.ru (B.T.S.); tlek84@mail.ru (T.M.M.); 2Research and Production Center of Microbiology and Virology, Almaty 050010, Kazakhstan; j.gulnar60@mail.ru; 3Faculty of Science, Department of Molecular Biology and Genetics, Mugla University, Mugla 48000, Turkey; rmammadov@yahoo.com; 4Department of Biology and Ecology, Faculty of Nature and Technology, University of Odlar Yurdu, AZ1072 Baku, Azerbaijan

**Keywords:** in vitro adventitious roots culture, *Allochrusa gypsophiloides*, Turkestan soap root, saponins, antifungal activity

## Abstract

*Allochrusa gypsophiloides* is a rare Central Asian species, a super-producer of triterpene saponins with pharmacological and technical value. In this work, a comparative evaluation of the in vitro culture of adventitious roots (ARs), in vitro adventitious microshoots (ASs), natural roots and aboveground parts of wild plants from Kazakhstan to define the total saponin (TS), phenol (TP) and flavonoid (TF) content, as well as antioxidant (AOA) and antimicrobial activity, is presented for the first time. In the AR culture, growth index (GI), TS, TP and TF were evaluated on days 25, 45 and 60 of cultivation on ½ MS medium without (control) and with auxin application. It was found out that TS and TF were higher in the in vitro AR culture. The amount of TP and TF are higher in the aerial part of vegetative plants with maximum AOA. The concentration of the extract required to inhibit 50% of 2,2-diphenyl-1-picrylhydrazyl (DPPH) radical formation (ICO_50_) in extracts from natural material negatively correlated with TS, TP, TF and in the in vitro AR culture with TF. Control extracts from the in vitro AR culture with high TS levels showed growth-inhibitory activity against *S. thermophillus*, *S. cerevisiae* and *C. albicans*. The influence shares of medium composition factor, cultivation duration factor and their interaction with GI, TS, TP and TF were determined. The in vitro AR culture is promising for obtaining triterpene saponins TSR with high antibacterial and antifungal activity, and the in vitro ASs culture—for shoot multiplication with antioxidant properties.

## 1. Introduction

*Allochrusa gypsophiloides* (Regel) Schischk (fam. Caryophyllaceae Juss.), Turkestan soap root (TSR), a rare species of the natural flora of Central Asia and Kazakhstan with a limited habitat in the Western Tien Shan and Pamir-Alai, is a valuable saponin-bearing plant [1,2]. The rare TSR species has pharmacological value as an expectorant, diuretic and a laxative natural remedy and technical application as an effective foaming agent and emulsifier [3]. TSR contains a complex of triterpene saponins of the oleanolic series, from which saponins—derivatives of gypsogenin and quillic acid—have been isolated [4,5]. They exhibit immunostimulant and antiviral properties [6]. High antioxidant activity and cytotoxic effect have been revealed in essential oils isolated from the flowering aerial parts of TSR plants [7]. Saponins are high-molecular steroidal or triterpene glycosides consisting of a hydrophilic carbohydrate part and a hydrophobic aglycone [8]. In plants, saponins as secondary metabolites perform a protective role against a wide range of pathogens and pests [9]. Plant saponins have high surfactant qualities and exhibit antibacterial, antioxidant, virucidal, anti-inflammatory, antifungal, antitumor, etc., activities [10,11]. The wide range in the activity of saponins is based on their high affinity for phospholipids of cell membranes, the ability to form insoluble complexes with sterols and proteins, lytic activity, etc. [12]. According to the literature review, the content of saponins varies significantly depending on the plant species and the age [13], accumulation organ and phenological phase of development [14] and the environmental and agronomic factors of growth [15]. The main phytochemical studies on TSR are extremely few and mostly conducted in the 60–90 s of the last century [4,16,17,18]. A recent article provides a detailed review of the ecology, phytochemistry, and biological activities of the genus *Allochrusa* [19]. In recent decades, there has been a significant decline in the abundance of TSR as a result of intensive management activities in the main habitats of the species [20]. In order to replace wild TSR raw materials, it is necessary to develop an alternative approach based on in vitro culture that allows us to obtain plant biomass with valuable metabolites regardless of external factors and to automate the process of secondary metabolite production in bioreactors [21,22]. Of particular interest are the cultures of adventitious roots (ARs) and shoots (ASs), which are characterized by genetic and biosynthetic stability [23]. Optimization of the main trophic and hormonal factors of the nutrient medium makes it possible to achieve intensive growth of plant biomass and a stable level of biologically active metabolite in the isolated culture [24,25]. Currently, AR cultures are used for the commercial production of a number of secondary metabolites described in the reviews [26,27,28]. AR cultures have been developed for such saponin-producing plants as *Saponaria officinalis* [29,30], *Gypsophila paniculata* [31], *Panax ginseng* [32,33] and others [34].

The aim of the research was to evaluate the content of saponins, phenols and flavonoids in *Allochrusa gypsophiloides* plant material obtained from natural populations (in situ), under conditions of introduction (ex situ), in in vitro culture of adventitious roots (ARs) and in vitro culture of adventitious shoots (ASs) and to compare their antimicrobial and antioxidant activities in model test systems.

## 2. Results and Discussion

### 2.1. Content of Secondary Metabolites in In Situ and Ex Situ Extracts

The content of secondary metabolites (SM) in the native material differed depending on the plant part, growth conditions and plant developmental phase. The quantitative predominance of saponins was found in both roots and aboveground parts of TSR plants (Table 1).

The TS content in roots from natural populations and ex situ roots did not differ from each other and averaged 13%. A significant excess of TP and TF in the natural roots compared to the roots from plants grown in the experimental field was revealed: TP by 2 times (t = 10.07), TF by 3.2 times (t = 14.31). These differences may be related to three factors: natural climatic growing conditions, the age of the original donor plants and age-related morphometric parameters of the root.

The influence of abiotic environmental factors in species habitats on the content of SM has been found for many species [35,36]. High temperature, solar insolation, drought, salinization and other environmental stress factors enhance the accumulation of SM to increase the resistance and survival of the species under extreme conditions [37,38].

The TSR growing conditions in the natural species habitats in the Turkestan region are characterized by a significantly dry climate with high heat availability, hot summers (up to 50 °C) and warm winters and the predominance of sandy sierozem soils. The average annual temperature is +50 °C, and the total precipitation does not exceed 125 mm per year. The natural conditions of TSR cultivation in the south-east of Kazakhstan (Almaty region) differed from the natural habitat by a sharper continental climate, higher moisture availability with annual precipitation up to 500 mm, less solar activity, a summer temperature not exceeding 34 °C and colder winter [39]. Consequently, it can be assumed that natural TSR plants accumulated more phenolic substances in response to stress conditions than ex situ plants introduced in milder climatic conditions.

The absence of significant differences in the TS level in in situ roots and ex situ roots may be due to age differences in the original plants and the influence of growing conditions. Our earlier work [40] with wild plants revealed that the saponin amount in the root varies with the age of the generative plant, the relative indicator of which is the root diameter. The amount of saponins was maximized in medium-sized roots (5 cm diameter) from middle-aged plant samples. Small roots from young plants that were four times smaller in diameter and had large roots with a diameter of 7 cm and above contained fewer saponins than middle-aged roots. It should be taken into account that at the end of vegetation, wild large roots with a raw weight of 1.75 kg and an average diameter of 9.5 cm and ex situ roots, which, in the eighth year of introduction, reached an average weight not higher than 0.3 kg and a diameter of 4 cm, were selected for the analysis in this work. This fact indicates that natural in situ roots were obtained from older generative plants, in which the content of saponins decreased during ontogenesis (Figure 1a).

The influence of the original TSR plant age and growing conditions on the saponin accumulation in the roots was noted in other studies. The saponin content in the roots of *Allochrusa gypsophiloides* (determined by gravimetric method) growing in Uzbekistan varied from 14.19 to 22.16% depending on their age [41]. Under the conditions of introduction, the saponin amount in the roots increased with the age of plants, from 9.5% in one-year-old to 23% in five-year-old TSR plants [16]. The maximum accumulation of saponins in roots (20%) was observed by the fifth year of cultivation in Turkmenistan [17].

The literature data on the influence of plant phase development on the accumulation of saponins for similar saponin-bearing plants growing in Eurasia are rather contradictory. It was found that the maximum amount of saponins is contained in the mature root at the flowering stage in the wound-healing ulcer *Anthyllis vulneraria* L. [42]. For medicinal soapberry *Saponaria officinalis* L., the vegetation phase did not significantly affect the content of saponins in the roots; with the increasing age of the plant, the amount of saponins decreased with a parallel increase in polysaccharides [43]. In the underground organs of Ferula hormones, the content of saponins detected by the spectrophotometric method was 7.27% [44]. A high content of 9.14% was found in rootstocks of Aralia manchurian growing in the Far East [45].

The content of saponins in the aerial part of the TSR natural plants varied greatly depending on the growth phase and external appearance (habitus) of the plants, which changed significantly during ontogenesis. During active shoot formation, the plant consisted of strongly deviated shoots 10–15 cm high with numerous suboppositely arranged linear green leaves. At the flowering stage, the plant looked like a spherical green bush 50–70 cm high and 60–100 cm in diameter with leaves and numerous panicles of whitish-pink small flowers (Figure 1b). At the end of the vegetation, the plant was a characteristic “tumble-weed” bush with dried shoots with numerous seed capsules at the top (Figure 1c).

In the aerial part, SM content varied depending on the phase of plant development. At the stage of active vegetation, the high content of TS exceeded their level in roots (t = 6.53), as well as their amount in flowering shoots and dry shoots at the seed stage (t = 4.37, t = 9.75). The maximum TP content in the aboveground part at the vegetation and flowering stages (3.5% and 4.2%, respectively) decreased sharply towards the end of vegetation at seed setting to 0.8%. The highest TF level up to 8.5% was found in flowering shrubs, in green shoots 3 times lower (2.9%), the minimum content was found at the end of vegetation, which is comparable to their level in native roots.

According to the literature data, in the aboveground part of the plants, saponins accumulate in the palisade tissue of leaves [14] and phenols and flavonoids in the tips and flowering plants [46]. In the TSR plants, the leaf cover is well-expressed during the vegetative and flowering stages, and by the end of the growing season, the aboveground part completely dries up and is represented by a dry bush with ripening seeds. The high metabolic profile of the above-ground part of TSR field plants was confirmed by the antioxidant activity of these extracts with the minimum ICO_50_ value (0.87 mg/mL) in this study (Table 1).

Thus, the SM content in the native TSR material varies significantly depending on the growing conditions and developmental phase of wild plants. The data obtained indicate the feasibility of growing *A. gypsophiloides* under introduction conditions in the south-east of Kazakhstan to create a TSR raw material base as a super-producer of triterpene saponins. The revealed high content of SM in the aerial part at the stage of vegetation and flowering makes it possible to alternatively harvest plant raw materials without digging up the roots to preserve the original plant and annual resumption of its growth.

### 2.2. In Vitro Adventitious Roots Culture

In this work, for the first time, a comparative assessment of the growth index (GI) and the content of three groups of SM in extracts from the in vitro AR culture of TSR obtained on control and experimental variants of the medium with the addition of different auxins was carried out (Figure 1d–f).

In the control medium, at the minimum culture duration of 25 days, the AR biomass yield did not exceed 1 g (DW 0.1 g) and GI 127. Increasing the culture time to 45 days resulted in an almost threefold accumulation of biomass up to 2.7 g and increased growth to GI 348. When the culture time was lengthened to 2 months, the biomass yield and AR gain decreased twofold, to 1.37 g and GI 180, respectively (Figure 2, Table A1).

On IBA medium, the pattern was similar to the control. At the same time, higher biomass and GI relative to the control were observed only on the 25th day of culture. When the duration of cultivation was increased up to 45 days, the growth indicators of ARs (FW 3.5 g, GI 458) did not differ from the control similar period. A further increase in the period up to 2 months on IBA medium led to a decrease in growth indicators to the level of the 25-day culture on the same medium and in the control of the same cultivation period. On NAA medium, the growth dynamics differed from the control and IBA medium. Thus, on the 25th day of culture, the biomass accumulation did not exceed the control and was significantly lower than that on the IBA medium. When increasing the culture period from 25 to 45 days, there was a slight increase in growth without a significant difference. When extending the culture period to two months, there was a significant increase in growth up to a maximum fresh weight of 4.5 g (DW 0.43 g) and GI 589, which exceeded the values of the other variants. The results of the one-factor analysis of variance (df 2, *p* < 0.05) showed a reliable influence of the cultivation duration factor on the culture growth index in all medium variants. At the same time, a higher dependence of growth on culture duration was observed on the medium with NAA (F = 3385, df 2, *p* < 0.001) compared to the IBA medium (F = 85, df 2, *p* < 0.001) and control (F = 37, df 2, *p* < 0.001).

It was found that saponins accumulate in predominant amounts in the in vitro AR culture (Figure 3, Table A2).

The TS content after 25 days of AR cultivation in the control and on NAA medium was comparable, at 11%, and on the IBA medium, did not exceed 8%. The control and experimental variants of ARs did not differ in the TF amount, and the TP level prevailed in the experimental variants and exceeded the control of a similar term.

A high TS amount of 27.7% was observed in the AR biomass on the 45th day of culture on the control medium, whereas on the NAA and IBA media, their levels were lower by 2 and 2.3 times, 14.6% and 12%, respectively. The TP content (1.4%) did not differ significantly between control and experimental variants. The TF amount in the control (1.58%) exceeded: 1.8 times in the IBA medium and 5 times in the NAA medium (0.9% and 0.3%, respectively). With increasing the duration of culture from 45 days to two months in the control, SM content in AR biomass decreased: saponins by 25%, phenols by 40% and flavonoids by 54%.

In the experimental variants, the content of SM varied depending on the auxin type in the medium. In the AR culture on the IBA medium, the increase in TS level was 47%, TP by 49%, and TF content remained at the level of 0.8%, which was inverse to the control. In the NAA medium, there was a slight increase in TS, the flavonoids level was preserved, but the TP amount decreased by 83% from 1.69% to 0.29%. The TS content in IBA medium and NAA medium for the second month of culture did not differ from each other and averaged 17.2%, which was lower than the control value of the same period.

The results of one-factor analysis of variance (df 2, *p* < 0.05) revealed a different degree of influence of the culture term on SM accumulation depending on the culture medium. A significant effect of the culture duration on TS accumulation (F = 582, df 2, *p* < 0.001) was found on the IBA medium. The TF level was significantly dependent on the duration of cultivation on the NAA-amended medium (F = 230, df 2, *p* < 0.001). The factor of cultivation duration did not significantly affect TF content on the medium with IBA (F < F_tabl_, df 2, *p* > 0.05) but had a significant effect on TF accumulation on the control medium (F = 111, df 2, *p* < 0.001).

In recent years, there has been an increasing interest in in vitro ARs for the production of species-specific secondary metabolites of many valuable rare plants. A recent review by Hussain et al. provides a comparative analysis of ARs and hairy roots culture studies with emphasis on their advantages and disadvantages for the commercial production of valuable metabolites. The advantages of AR culture include stability of growth and accumulation of SM in the extracellular space, which facilitates extraction, as well as safety and the high commercial value of the natural (untransformed) product obtained. The disadvantages of AR culture include high cell water content, foaming, cell wall growth and relatively unstable production in bioreactors [24].

Currently, an in vitro AR culture is being developed to produce a wide range of SM. The optimal conditions for mass proliferation of in vitro ARs of *Hibiscus hamabo* in a bioreactor for the production of phenols and flavonoids have been developed [47]. The in vitro AR culture of *Rheum emodi* was found to be promising as an alternative source for the production of anthraquinones with anticancer effects [48]. The in vitro AR culture of *Primula veris* sursp. *veris* showed a 5.67 times higher amount of primulic acid II than in soil-grown roots, but primulic acid I content, on the contrary, was 3 times higher in soil-grown roots [34].

In an in vitro culture, inducers of adventitious root differentiation are phytohormones or imitators of their action and growth regulators, which trigger early physiological processes that ensure the development of root primordia [23,24]. The key role of auxins in the establishment and formation of adventitious roots has been established, and the effectiveness of different auxins on the induction and proliferation of ARs differs depending on the original species and cultivation conditions [49,50]. Therefore, the effect of the hormonal composition of the nutrient medium on differentiation, growth intensity and accumulation of root biomass and yield of bioactive metabolites should be studied for each culture.

The growth-stimulating effect of auxins on AR biomass increase has been established for different species. A high saponin level of 29.5 mg/g crude weight was detected in a *Primula veris* root culture on an inducing medium with auxin in combination with kinetin [51]. In the in vitro culture of *Gypsophilla* ssp. roots, the accumulation of saponins varied from a minimum, 7 mg/g, to a maximum level, 65 mg/g, depending on the original root line [31]. In the work of M. Simao et al. [49], the effectiveness of NAA action on the proliferative activity of adventitious roots of *Passifflora pohlii* Mast is shown.

A number of studies revealed that the duration of cultivation on the inducing medium also affects the content of secondary metabolites and their qualitative composition in the in vitro AR culture. The dependence of tropane alkaloid and phenol accumulation on the duration of cultivation and L-phenylalanine treatment in the in vitro AR culture of *Hyoscyamus niger* L. was shown [52]. The effect of the culture period on the yield of secondary metabolites with an optimal 15-day period is shown in the in vitro AR of *Hyoscyamus niger* L. [53]. The HPLC of the in vitro culture of Lang Bian ginseng showed that only twenty-week-old in vitro rhizomes had ginsenoside Rg1, Rd and Rb1 [54].

In this work, the comparative evaluation of the content of three SM groups in the in vitro AR culture TSR was conducted for the first time. Our data showed a quantitative excess of saponins and flavonoids in the in vitro AR culture compared with native roots. At the same time, nutrient medium factor, cultivation duration factor and their joint interaction had a significant effect on the quantitative growth indicator of the in vitro AR culture and the content of SM in it.

To reveal the relationship between AR biomass growth and accumulation of secondary metabolites during cultivation on control and auxin-containing medium variants, a correlation analysis was performed (Figure 4).

On the control medium, the content of SM increased with the increasing GI and decreased with its decrease. A high correlation coefficient (CC) at *p* < 0.05 was found for saponins (CC +0.84) and for flavonoids (CC +0.94). The TP content in the control culture did not significantly depend on GI.

In the experimental variants, the nature of the relationship between GI and SM accumulation differed both from the control and among themselves. In the variant with NAA, as in the control, there was a positive correlation between the accumulation of saponins and growth (CC +0.76). At the same time, there was a significant negative correlation (CC −0.99, *p* < 0.05) between growth and the phenol level, which steadily decreased from 1.6% to 0.3% by the end of cultivation.

No significant correlation between growth and SM content was found on the IBA medium. A twofold increase in GI from 25 to 45 days of culture did not change the TP and TF levels. The maximum TP content 2.12% was observed at 60 days with a low GI 212 (CC −0.46). TF amount was maintained at the same level during cultivation (CC +0.64). The TS amount increased during cultivation from 7.96% to 17.64%, regardless of the drop in growth at the end of cultivation.

A two-factor variance analysis of the data revealed the influence of the duration of cultivation and hormonal composition of the nutrient medium on the biomass growth of ARs and the SM accumulation in them (Table A3, Figure 5).

It was found that the in vitro AR culture growth was more dependent on culture duration than on the medium composition (27.6% and 4%, respectively). However, the interaction of the two factors had the maximum effect on GI with a share of 66.5%. The effect of the cultivation period on saponin accumulation was more significant than that of the medium (45% and 27%, respectively). The duration of cultivation did not affect TP and TF levels (13%). The effect of the nutrient medium on the TP amount was significantly enhanced when it interacted with the factor of culture duration with an increase in the proportion of influence up to 60%. The TF content in the AR culture was significantly influenced by the composition of the nutrient medium and, to a lesser extent, by the combined effect of the two factors.

Based on the obtained data, it can be assumed that the control medium is preferable for the joint accumulation of saponins and flavonoids in the AR culture at the optimal cultivation period of 45 days, the medium with NAA—for obtaining an in vitro AR culture with a high content of saponins and a low level of phenols at two-month cultivation, the medium with IBA—for an in vitro AR culture with a high content of saponins and phenols at two-month cultivation.

### 2.3. In Vitro Adventitious Shoot Culture

Adventitious shoots in vitro arise as a result of bud regeneration on nodal shoot explants, i.e., without callus formation, by direct organogenesis. Direct regeneration in vitro is regulated by the growth regulators cytokinins supplemented in the nutrient medium, which activate the growth of buds and shoot development from them. Micropropagation by adventitious shooting is used for the multiplication of rare plants and isolation of valuable substances from them without damaging natural populations [55,56].

The study results of a comparative evaluation of the in vitro metabolic profiles of plant material and natural or field plants are contradictory. A quantitative excess of flavonoids and anthocyanins in the in vitro micro clones regarding wild plants was found for the endemic *Saussured costus* [57]. In the in vitro shoot cultures of *Lychnis flos-cuculi* L., the content of phenols was comparable to their level in the flowering herb collected from natural sites, and the flavonoids were lower. The high content of polyphenols correlated with the AOA of the extracts [46]. The amount of quercetin in root extracts from in vitro shoots of *Nardostachys jatamansi* and their AOA did not differ from root extracts from wild plants [58]. A comparative HPLC analysis of in vitro plants and intact plants of *Spiraea betulifolia* revealed: in regenerants—a wider range of phenolic components and a twofold higher concentration of chlorogenic acid and kaempferol, and in intact plants—increased content of hyperoside, astragalin and quercetin [59]. The field-grown plants of *Astragalus gymnolobus* showed higher amounts of phenolic compounds and higher antibacterial and antioxidant properties than the in vitro-regenerated plants [60].

It was previously shown that TSR has a high regenerative potential in vitro, which makes it possible to use the micropropagation for the preservation and multiplication of a rare species [61]. In this work, for the first time, the content of SM in aseptic TSR shoots replicated by microcutting (Figure 1g,h) is evaluated. It was found that microshoots contained fewer SM than wild plants (Table 1 and Table A2). The level of saponins in microshoots was two times lower than that in natural plants. Compared with native vegetative shoots (similar in morphology), the TP and TF content in microshoots was two or more times lower, and regarding the dry shoots at the end of vegetation, on the contrary, the amount of phenolic substances was higher in microshoots. Aseptic microshoots compared with in vitro ARs contained less TS but more TP and TF, as did the aboveground part of the wild plants. The predominant TS content in ASs did not exceed 7.7%, which was significantly lower than in the 45-day and two-month-old in vitro AR cultures.

Thus, the in vAS culture of *A. gypsophiloides* is of interest for obtaining and further studying phenolic substances of a rare species. An important advantage of in vitro AS cultures is the possibility of year-round obtaining plant material in mass quantities in a shorter time than in field conditions.

### 2.4. Antioxidant Activity of Extracts

The AOA on the inhibition of DPPH radical formation in TSR extracts was generally low and amounted to a concentration of 1 mg: 56% maximum and 10% minimum. The ICO_50_ value varied depending on the part and developmental phase of the field plants (Figure 6, Table 1).

The highest AOA with the minimum ICO_50_ (0.87 mg/mL) was found in the extract from the aerial part at the vegetation stage, which did not change significantly in the flowering plant. At the end of vegetation, the AOA of extracts from the aboveground part decreased almost twofold, to ICO_50_ 2.18 mg/mL, which is comparable to the ICO_50_ of extracts of in situ and ex situ roots.

The AOA activity in the in vitro AS extracts was lower than that of native shoots at the vegetative and flowering stages. The ICO_50_ of aseptic shoots did not differ significantly from the ICO_50_ value in extracts from the aerial part at the end of vegetation and extracts from native roots. In the in vitro culture, the minimum ICO_50_ value was 1.07 mg/mL in the AR extract on 45-day control medium, which did not change significantly with the time prolongation of cultivation. The experimental extracts on the NAA medium had a significantly higher ICO_50_ value than the control at all culture durations in contrast to the extracts on the IBA medium with an ICO_50_ value at the control level.

The literature review showed significant differences in the AOA and ABA of extracts from in vivo and in vitro plant material depending on their phenolic content. Increased phenolic content and high AOA was observed in extracts from the leaves of field plants compared to in vitro plants of *Passiflora alata* [62]. The AOA of the extracts from the primary and secondary roots of in vivo-grown plants of *Passiflora pohlii* Mast was higher than the AOA of root extracts from in vitro-grown plants [49]. High levels of polyphenols in the in vitro shoot cultures of *Lychnis flos-cuculi* L. correlated with the AOA of the extracts [46]. The field-grown plants of *Astragalus gymnolobus* showed higher amounts of phenolic compounds and higher antibacterial and antioxidant properties than the in vitro-regenerated plants [60].

The correlation analysis on the AOA of extracts (ICO_50_) and the content of SM in the extracts showed a different nature of the relationship between the indicators in native raw materials and the in vitro AR culture (Figure 7).

It was found that the AOA of extracts from the natural plant material depended on the amount of all three SM groups. It was found that the ICO_50_ value of natural extracts was more strongly negatively correlated with TS level (CC −0.82) than the levels of TP and TF, but no significant correlation was revealed (*p* < 0.05). The higher the TS, TP and TF content, the lower the ICO_50_ index and, consequently, the higher the antioxidant properties of the extracts. The highest AOA with minimum ICO_50_ (0.87 mg/mL) was found in the extract from the aerial part during the vegetation phase, which increased to ICO_50_ 1.2 in the flowering phase (Table A2). It was during these developmental periods that the plant contained the maximum levels of phenolic substances, flavonoids and saponins.

AOA in the in vitro AR culture was determined to a greater extent by the content of TP than by the TS and TP amounts in the AR biomass. There was a significant negative correlation (CC −0.89) between TF content and ICO_50_. The lowest ICO_50_ with a value of 1.07 was found in the control extract on day 45 of the culture with a maximum flavonoid content of 1.58% (Table A2). The positive correlation between TP and TF and AOA and the crucial role of flavonoids in antioxidant potential were established using multivariate statistical analysis for four *Thyme* species [63].

### 2.5. Antibacterial and Antifungal Activities of TSR Extracts

The antimicrobial activity of TSR plant material was assessed for the first time in this work. The testing of extracts from native roots and aerial parts of field plants did not reveal any inhibitory activity against the used bacteria at all tested doses. However, the extracts obtained from the in vitro AR culture showed an antagonistic effect against Gram-positive and Gram-negative bacteria (Table 2).

The zone of action of the antibiotic lincomycin on the growth of *P. aeruginosa* exceeded the experimental zones by 2–3 times. The control extract from 1.5-month AR culture at 25 μL inhibited the growth of *P. aeruginosa* with a zone of 5 mm (Figure 8a).

The extracts from 1.5-month-old AR culture on the NAA medium showed no activity against any bacterium at all the tested doses. The control extracts from 1.5-month and 2-month AR cultures significantly inhibited the growth of Gram (+) *S. thermophillus* at 100 μL (Figure 8b). At the same time, no sensitivity of *S. thermophillus* to lincomycin was detected at 20 μL.

All extracts from the two-month-old AR culture showed growth-inhibitory activity against *B. subtillis* at a dose of 250 μL, which was comparable to the effect of the antibiotic lincomycin at a dose of 20 μL. The commercial saponin from *Quillaja bark* at a dose of 25 µL did not show inhibitory properties against any of the tested bacteria.

The antifungal activity of the in vitro TSR extracts against *C. albicans* and *S. cerevisiae* was revealed (Table 3).

The negative control, 70% ethanol at a dose of 20 μL, had a negative effect on *S. cerevisiae* with a zone of action of 4 mm but was not effective against *C. albicans*. The antifungal drug fluconazole, 20 μL, inhibited the growth of both test subjects with inhibitory zones of 9 mm and 5 mm, respectively. Of the tested extracts, only the control extracts showed antagonistic activity against *S. cerevisiae* at a dose of 100 μL and 250 μL with an average zone of action of 4.7 mm. Against *C. albicans*, the inhibitory activity of all extracts was shown when a concentrated dose (10:1) containing 1.0 mL or 1.25 mL of the extract was applied to the disk. An ethanol solution of commercial saponin from *Quillaja bark* (1:1) showed high inhibitory activity with inhibition zones of 14.8 mm and 16.2 mm, respectively, at a dose of 25 µL/disc (Figure 8c). The zones of action of the one-and-a-half-month control and the experimental extracts were lower than that of the saponin from *Q. bark* but significantly higher than that of fluconazole.

The antifungal activity of saponin extracts has been identified in many studies and described in reviews [9,11,12,64]. A high efficacy against strawberry phytopathogens was found for saponins isolated from a suspension culture of *Solanum chrysotrichum* [65]. Triterpene saponins isolated from *Sapindus mukorossi* pulp extracts showed inhibitory activity against *Trichophyton rubrum* and *C. albicans* with an MIC_80_ value of 8 μg/mL [66]. The growth-inhibitory activity of saponin extracts from the roots of *Glycyrrhiza glabra* and *Quillaja saponaria* against *C. albicans*, with an inhibition zone of 15.3 and 16.5 mm, was evident at a dose of 20 mg/mL [67]. In our experiments, the zone of inhibition of TSR extracts from the in vitro AR culture was 10.7 mm at a much lower dose (1.25 mg/mL).

Significant antimicrobial activity of the in vitro AR culture extracts was revealed, which was absent in extracts of the native roots, aboveground parts of field plants and in vitro shoots. High antibacterial and antifungal activities were possessed by the in vitro AR culture extracts obtained on the control medium, characterized by the maximum amount of saponins.

## 3. Materials and Methods

### 3.1. Plant Material In Situ and Ex Situ

The initial plant and seed material of wild *Allochrusa gypsophiloides* were collected from natural populations in the desert zone in the south of Kazakhstan in 2022. The experimental plants were collected on an elevated undulating plain at an altitude of 375 to 670 m above sea level. The roots were excavated from model specimens of generative plants at the flowering–fruiting phase (August). The aboveground part of the plant was sampled in the active phase of vegetation, flowering–fruiting and at the end of vegetation at the seed stage “tumble-weed”. The plant material of *A. gypsophiloides* was identified at the Institute of Botany and Phytotroduction of MES RK. Further storage of the material was carried out in parchment bags at room temperature and 40% humidity. Each analyzed group included plant material from roots or aboveground parts from three plants at the same vegetation phase, having close root morphometric parameters: length, 25–30 cm; raw weight, 1.2–1.5 kg; diameter, 8–11 cm. The seed material collected during the expeditionary survey of the natural populations in the Turkestan region [68] was used for TSR introduction ex situ. Seed sowing was preliminarily carried out in the fall of 2014 at the experimental site in the southeast of Kazakhstan (Almaty region). For analysis, root excavation was carried out in the 8th year of introduction at the flowering–fruiting stage. The roots had an average diameter of 4 cm and a fresh weight of 0.3 kg.

### 3.2. In Vitro Culture Plant Material

In vitro establishment and micropropagation of the initial plants of *A. gypsophiloides* were carried out according to the technique described in our previous work [61]. Nodal segments of shoots from vegetative field plants were used as explants. The basic nutrient medium was Murashige and Skoog (MS), supplemented with 1 mg/L kinetin and 0.5 mg/L NAA. Cultivation conditions: 16-h photoperiod, temperature of 22–24 °C, humidity of 60% and illumination of 3000 lux. The cultivation cycle on the inducing MS medium was 30 days with further microcutting and double passaging on a similar medium; subsequent passages were carried out on the medium without growth regulators. Seeds of natural populations were deposited under liquid nitrogen conditions to preserve germination [68] and were used as the starting material for isolated root cultures. Seeds were pretreated with 0.01% gibberellic acid solution for 2 h and then transferred to Knop’s nutrient medium to obtain aseptic seedlings under stationary conditions in the light room. Obtaining the in vitro AR culture was carried out according to a previously described technique [69]. Tips of the main and lateral roots were isolated from aseptic two-week-old seedlings and cultured in 100-mL flasks containing 50 mL of liquid medium on orbital shakers (IKA, Staufen, Germany) at 100 rpm at room temperature under artificial shade conditions. An MS basic liquid nutrient medium with half the concentration of macro- and micro-salts (½ MS) without growth regulators was used as an in vitro control, while the experimental variants were ½ MS with the addition of auxins at a concentration of 1 mg/L, indolyl-3-butyric acid (IBA) or α-naphthyl acetic acid (NAA). Dry nutrient mixtures and growth regulators produced by Sigma-Aldrich (St. Louis, MO, USA) were used. Ten root explants with a total average fresh weight of 7.6 mg were inoculated into the culture flask. The repetition of the experiments was 8–10 flasks in each variant. The duration of cultivation was 25, 45 and 60 days, and at the end of each incubation period, the material was sampled to determine the fresh (FW) and dry weight (DW) of the AR biomass. The obtained average biomass data were expressed in g/flask containing initially 50 mL of nutrient medium. The root biomass growth index (GI) was calculated as: (harvested fresh weight − initial fresh weight)/initial fresh weight.

### 3.3. Obtaining Total Extractions

To obtain the total ethanol extracts, the standard methods with some modifications were used [70]. Dry wild plant materials (2–5 g) were preliminarily purified from undesirable impurities by chloroform, followed by extraction with 70% ethanol (1:100) for 2 h in a Sosklet apparatus, and then, ethanol extractions were distilled under vacuum to a dry residue. To obtain the total extracts from the in vitro AR culture, dry root biomasses (1–2 g) obtained on the 25th, 45th, or 60th day of cultivation on the control and experimental variants of ½ MS medium were extracted with 70% ethanol, followed by distillation on a rotary evaporator. The total extracts from in vitro ASs were obtained at the end of one month of cultivation on the MS medium without growth regulators. All obtained extracts were lyophilized and stored at −20 °C until analysis.

### 3.4. Quantitative Determination of Secondary Metabolites

The total content of saponins (TS), phenols (TP) and flavonoids (TF) was determined in ethanolic extracts: (1) from wild plant material from natural populations in situ; (2) from roots under introduction ex situ; (3) from the in vitro culture of adventitious roots (ARs) and of adventitious shoots (ASs). The TS content was determined by a spectrophotometric method based on the specific reaction of vanillin and sulfuric acid with the OH group at the C_3_ atom in both free and glycosylated forms of saponins [71,72]. The optical density of the solutions was measured at 544 nm, and the results in terms of oleanolic acid standard by concentration curve (y = 4.9976x + 0.0019, R^2^ = 0.9876) were expressed in *w*/*w*% using formula:

TS = C × V_t_ × D × 10^−3^ × 100/V_al_ × m_s_
(1)

where TS—total saponin content (% *w*/*w*), C—concentration from the curve (mg/mL), V_t_—total volume of extract (mL), V_al_—volume of aliquot (mL), D—dilution factor, m_s_—the mass of extract (g), 10^−3^—conversion factor for mg in g, 100 − %.

The quantification of TP in the extracts was carried out by the spectrophotometric method with the Folin–Ciocalteu reagent using gallic acid as standard (y = 0.9363x − 0.0008, R^2^ = 0.9948) at of 760 nm [73]. The spectrophotometric method based on the reaction of complex formation with aluminum chloride was used for the estimation of TF [74]. The results of the analysis were expressed in terms of quercetin by concentration curve (y = 4.708x − 0.0044, R^2^ = 0.9995) at 415 nm. TP and TF were calculated using a formula similar to saponins.

### 3.5. DPPH Radical Scavenging Assay

The antioxidant activity of the extracts was evaluated using the DPPH method by Sanja et al. [75]. Stock solutions of the extracts, prepared at 1 mg/mL concentrations by dissolving in methanol, and various concentrations (200–1000 mg/mL) were tested. Ascorbic acid was used as a positive control. The DPPH radical-scavenging activity was calculated using the following equation: % AOA = (A_o_ − A_1_)/A_o_ × 100, where A_o_ is the absorbance of the control, and A_1_ is the absorbance of the extract. A percent inhibition versus concentration curve was plotted, and the amount of antioxidants necessary to decrease the initial DPPH concentration by 50% (ICO_50_ value) was calculated.

### 3.6. Antimicrobial Activity of Total Extracts

The antimicrobial activity of the total extracts was determined by the method of diffusion into agar using paper disks [76] on the basis of the laboratory of agricultural and environmental microbiology of Scientific and Production Center of Microbiology and Virology, Kazakhstan, Almaty. Collection strains of Gram-positive bacteria *Bacillus subtillis*, *Streptococcus thermophillus*, Gram-negative bacteria *Escherichia coli*, *Pseudomonas aeruginosa*, yeast *Saccharomyces cerevisiae* and *Candida albicans* were used as test-cultures. To evaluate the antimicrobial activity, the total extracts were dissolved in 70% ethanol in a 1:1 ratio and 25 µL, 50 µL, 100 µL and 250 µL were applied on a disk. For the test culture of *C. albicans*, concentrated extracts were prepared in a ratio of (10:1), and 100 µL, 125 µL and 250 µL were applied on a disk with the corresponding initial extract contents of 1 mg, 1.25 mg and 2.5 mg. On the surface of the nutrient medium fish-peptone agar (FPA), MRS (for bacteria) or Sabouraud (for yeast), 0.1 mL of test culture was dispersed using a Drigalsky spatula and the extract-impregnated disks were laid out. The obtained cultures were incubated in the thermostat for 24 h at 37 °C. As a negative control, 70% ethanol at a dose of 20 μL was used. The antibiotic lincomycin for bacteria and fluconazole for yeast at 20 μL/disc were used as positive control. Commercial saponin from *Quillaja bark* (Cas N:8047-15-2, Sigma, St. Louis, MO, USA) at a concentration of 25 μL/disc (stock solution 1:1) was used as a comparison drug. The antagonistic activity was judged by the diameter of the zones of no pathogen growth formed around the disks using a ruler-tool to measure the zone of inhibition.

### 3.7. Statistical Analysis

All the experiments were performed in three biological and three technical replicates. Atypical values were excluded from the data based on *t*-tests, and the standard error of the average sample was calculated using the parametric Student’s test at *p* < 0.05 according to State Pharmacopoeia [77]. The results of the experiments were expressed in tables as means ± SE (standard error) and in charts as means ± CI (confidence interval) by *t*-test, *p* = 0.05.

The processing of the data and graphing were performed using Microsoft Excel (Microsoft Corp., Redmond, Washington, DC, USA). The one-way analysis (ANOVA) for growth index (GI), total saponins (TS), phenols (TP) and flavonoids (TF) in in vitro AR culture on hormone-free (control) and auxin-containing ½ MS medium was conducted with the Statistica software system (StatSoft Inc., Tulsa, OK, USA, 2011 version 10). The influence of the cultivation duration factor, the medium composition factor and their combined effect on GI and the total content of TS, TP and TF were evaluated by two-factor analysis of variance (ANOVA test) by Statistica software. A person correlation test was performed to establish significant effects among variables: GI and TS, TP, TF; AOA (DPPH assay) and TS, TP, TF. The differences were considered significant at *p* < 0.05.

## 4. Conclusions

The conducted studies revealed differences in the content of secondary metabolites in in vitro culture material and in wild plants, as well as in their AOA and antimicrobial activity in model test systems. The amounts of phenolics and flavonoids were higher in wild plants at the vegetative-flowering stage. Aseptic microshoots were superior to in vitro AR culture in the amount of phenolic substances and did not differ in AOA from natural vegetative plants. The levels of saponins and flavonoids were higher in AR culture at optimal timing and medium of cultivation compared to native roots. High ABA and antifungal activity were detected in control extracts of in vitro AR culture with increased levels of saponins. The hormonal composition of the medium determined the nature of the relationship between AR culture growth and the accumulation of secondary metabolites in it. The content of saponins and flavonoids positively correlated with the growth of the culture on the control medium and on the NAA medium. By selecting the optimal hormonal composition and duration of cultivation, it is possible to obtain AR cultures with a quantitative predominance of saponins and with different levels of phenols and flavonoids.

## Figures and Tables

**Figure 1 plants-12-03521-f001:**
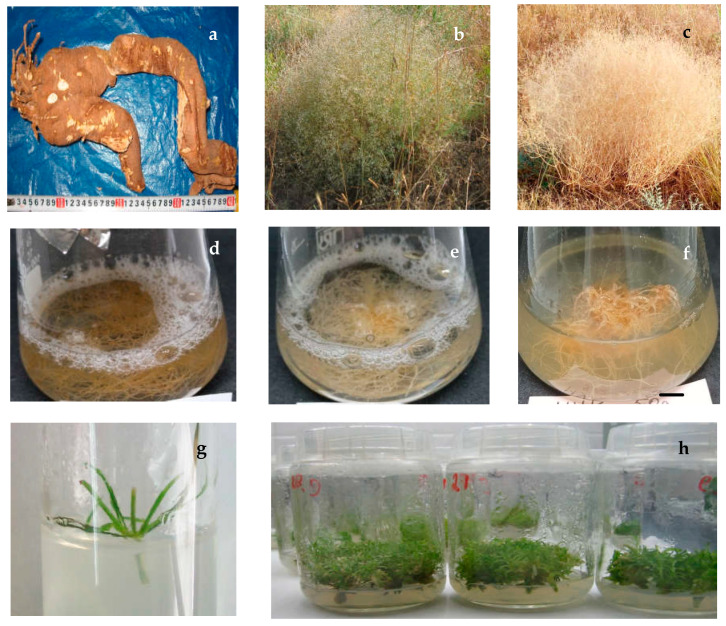
*Allochrusa gypsophiloides*: (**a**) wild-grown root from native site; (**b**) wild flowering plant; (**c**) bust “tumble-weed”; (**d**) in vitro adventitious root (AR) culture in liquid ½ Murashige and Skoog medium (MS) free from growth regulators (control) 45th day of culture; (**e**) in vitro AR culture in ½ MS with indolyl-3-butyric acid (IBA) 1 mg/L 45th day of culture; (**f**) in vitro AR culture in ½ MS with α-naphthalene acetic acid (NAA) 1 mg/L 45th day of culture; (**g**) initial node explant on MS medium with 1 mg/L kinetin and 0.5 mg/L NAA, 5th day of culture; (**h**) in vitro adventitious shoots (ASs) on MS without growth regulators, a month culture, 2nd passage.

**Figure 2 plants-12-03521-f002:**
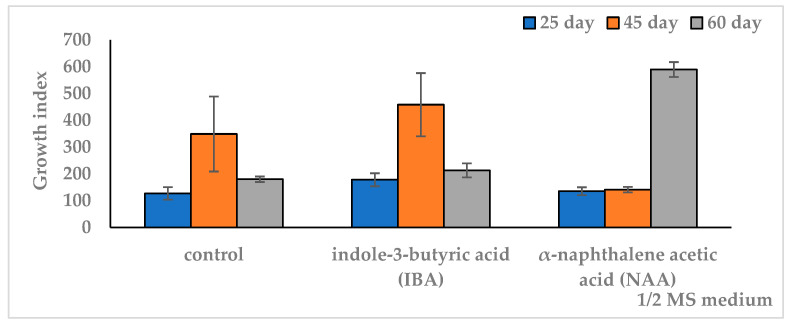
Effect of auxins in ½ MS nutrient medium on growth index (GI) of in vitro adventitious root (AR) culture of *Allochrusa gypsophiloides* during the two-month cultivation. ½ MS medium free from auxins (control). IBA indole-3-butyric acid, NAA α-naphthalene acetic acid at a concentration of 1 mg/L. The data are presented as means and their confidence intervals M ± CI by *t*-test (*p* = 0.05, n = 3). Bars indicate confidence intervals.

**Figure 3 plants-12-03521-f003:**
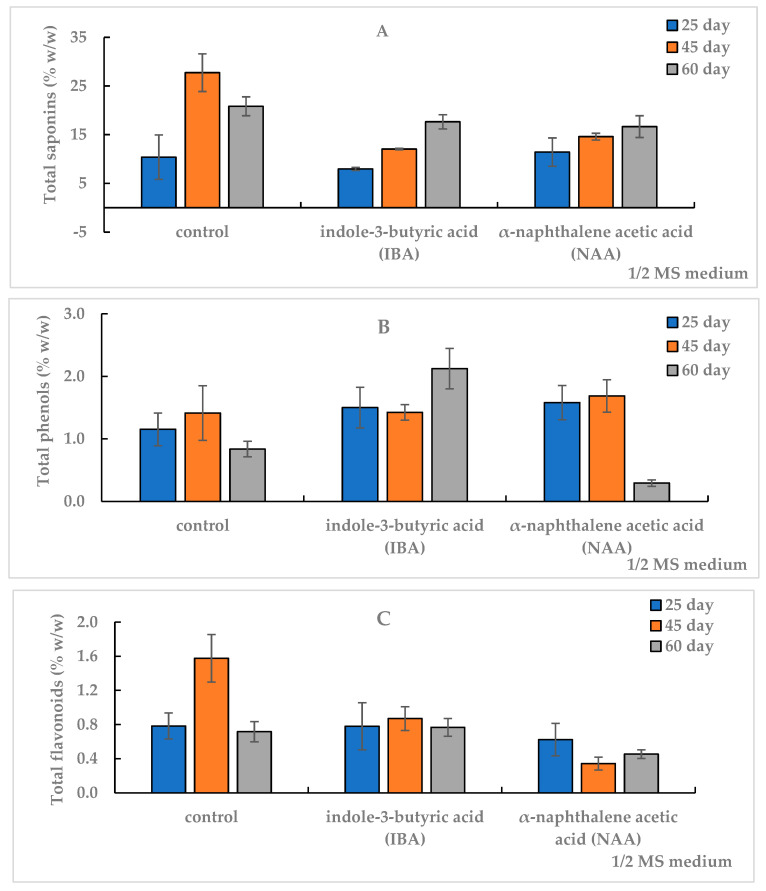
Total content of saponins (TS) (**A**), phenols (TP) (**B**) and flavonoids (TF) (**C**) in the in vitro adventitious root (AR) culture of *Allochrusa gypsophiloides* on the 25th, 45th and 60th days of cultivation on ½ MS medium free from auxins (control) and experimental variants with indole-3-butyric acid (IBA) or α-naphthalene acetic acid (NAA) at 1 mg/L. Means and their confidence intervals M ± CI by *t*-test (*p* = 0.05, n = 3). Bars indicate confidence intervals.

**Figure 4 plants-12-03521-f004:**
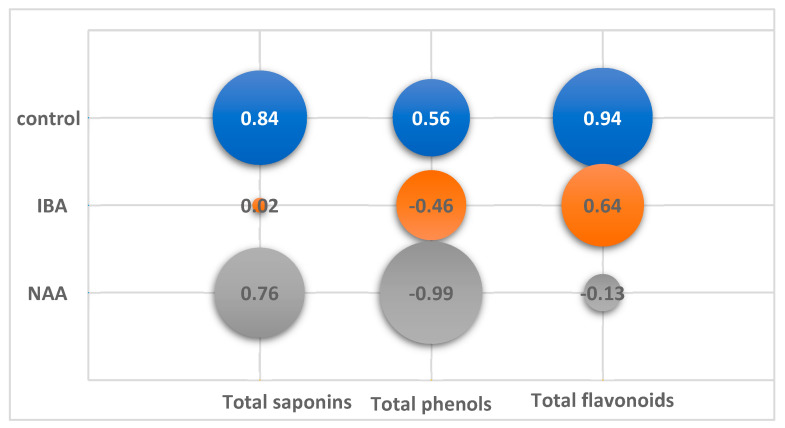
Correlation of total content of saponins (TS), phenols (TP) and flavonoids (TF) from growth index (GI) of in vitro adventitious root (AR) culture of *Allochrusa gypsophiloides* during two mounths of cultivation on control and experimental variants of ½ MS medium with indole-3-butyric acid (IBA), α-naphthalene acetic acid (NAA) at a concentration of 1 mg/L.

**Figure 5 plants-12-03521-f005:**
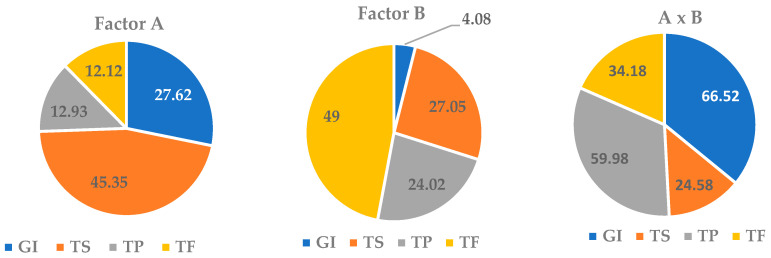
Shares of the influence of the cultivation duration factor (A), the nutrient medium composition factor (B) and their interaction (A x B) on the growth index (GI) and the total content of saponins (TS), phenols (TP) and flavonoids (TF) in the in vitro culture of adventitious roots (ARs) of *Allochrusa gypsophiloides*.

**Figure 6 plants-12-03521-f006:**
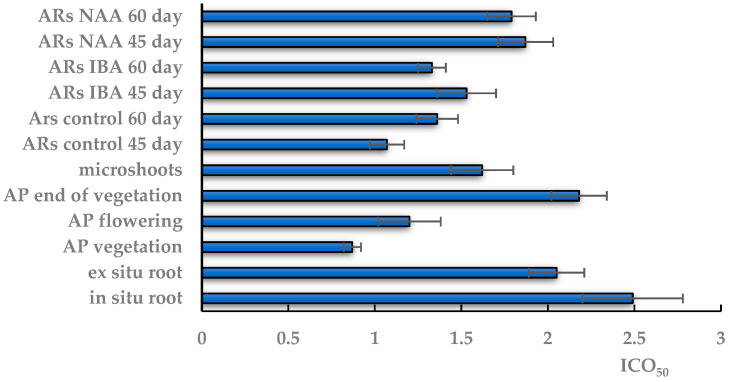
Antioxidant activity of TSR extracts obtained from in situ and ex situ roots; the aerial part (AP) of wild-grown plants during vegetation, flowering and at the end of vegetation; in vitro microshoots; in vitro adventitious root (AR) culture on the 45th and 60th days of cultivation on control and experimental variants of ½ MS medium with indole-3-butyric acid (IBA) or α-naphthalene acetic acid (NAA). ICO_50_—the concentration required to inhibit 50% of the DPPH radical formation. The data are presented as means ± standard error (SE).

**Figure 7 plants-12-03521-f007:**
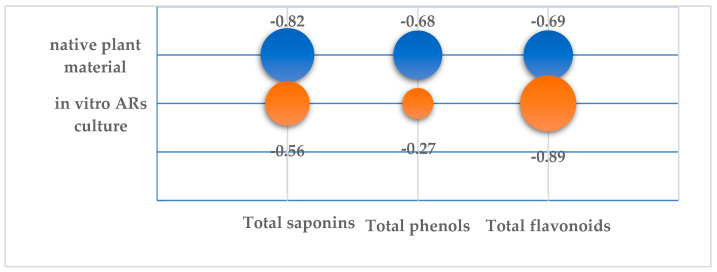
Correlation coefficients of ICO_50_ value of the TSR extracts from native plant material and in vitro AR culture with the total content of saponins (TS), phenols (TP) and flavonoids (TF).

**Figure 8 plants-12-03521-f008:**
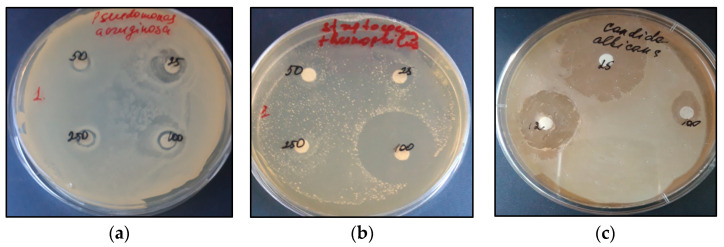
Growth-inhibiting activity of the control extracts of an in vitro AR culture of TSR in relation to: *P. aeruginosa*, zone 5 mm at 25 μL (**a**); *S. thermophilus*, 18 mm at 100 µL (**b**); inhibitory zone of 16.2 mm of commercial saponin from *Quillaja bark* (1:1) at 25 µL in a culture of *C. albicans* (**c**).

**Table 1 plants-12-03521-t001:** Content of total saponins (TS), phenols (TP) and flavonoids (TF) in ethanol extracts from native roots and aboveground parts of *Allochrusa gypsophiloides* and their antioxidant activity (AOA) by DPPH assay.

Part of the Plant	Conditions, Vegetation Phase	Total Content, % *w*/*w*			AOA	
		TS	TP	TF	%	ICO_50_ (mg/mL)
Root	in situ *	13.55 ± 0.40 ^ab^	2.83 ± 0.08 ^a^	0.46 ± 0.01 ^a^	18.47	2.46 ± 0.29 ^a^
	ex situ **	12.63 ± 0.48 ^ab^	1.32 ± 0.06 ^b^	0.14 ± 0.01 ^b^	13.23	2.05 ± 0.16 ^a^
Aerial	vegetation	16.50 ± 0.21 ^c^	3.54 ± 0.16 ^c^	2.85 ± 0.11 ^c^	55.79	0.87 ± 0.05 ^b^
	flowering	14.21 ± 0.48 ^a^	4.17 ± 0.11 ^c^	8.45 ± 0.31 ^d^	42.28	1.20 ± 0.18 ^bc^
	end of vegetation (“tumble-weed”)	12.01 ± 0.41 ^b^	0.79 ± 0.04 ^d^	0.53 ± 0.03 ^a^	11.66	2.18 ± 0.16 ^ad^

* Fresh weight—1.75 kg, diameter—9.5 cm, ** fresh weight—0.3 kg, diameter—4 cm. AOA (%) against DPPH radicals at a concentration 1 mg/mL, ICO_50_—the concentration required to inhibit 50% of DPPH radical formation. The data are presented as means ± SE of three independent experiments. Significant differences in the same column are represented by different letters (a–d), *p* ≤ 0.05.

**Table 2 plants-12-03521-t002:** Antibacterial activity of ethanol extracts of in vitro adventitious root (AR) culture of *Allochrusa gypsophiloides*.

Culture and Control	½ MS	Mean Diameter of Inhibition Zones (mm)						
		µL	*B. subtillis*	µL	*S. thermophillus*	*E. coli*	µL	*P. aeruginosa*
ARs 45 day	control	-		100	18.17 ± 0.76 ^a^	-	25	4.87 ± 0.23 ^a^
	NAA	-		-		-	-	
	IBA	-		100	5.77 ± 0.25 ^b^	-	-	
ARs 60 day	control	250	4.63 ± 0.15 ^ac^	100	13.33 ± 0.58 ^c^	-	250	3.80 ± 0.20 ^b^
	NAA	250	4.57 ± 0.25 ^ac^	-		-	-	
	IBA	250	3.97 ± 0.21 ^a^	-		-	-	
Ethanol		20	-	2.10 ± 0.10 ^d^		-	3.00 ± 0.1 °C	
Lincomycin		20	5.0 ± 0.1 °C	-		5.0 ± 0.10 ^b^	9.60 ± 0.45 ^d^	
Saponin of *Quillaja bark*		25	-	-		-	-	

- No activity; in vitro adventitious root (AR) culture, 45- and 60-day, cultivation period, control ½ MS without auxin, IBA indole-3-butyric acid, NAA α-naphthalene acetic acid at a concentration of 1 mg/L. The data are presented as M ± SE. Significant differences in the same column are represented by different letters (a–d) *p* = 0.05.

**Table 3 plants-12-03521-t003:** Antifungal activity of ethanol extracts of in vitro AR culture of *Allochrusa gypsophiloides*.

Culture and Control	½ MS	Mean Diameter of Inhibition Zones (mm)			
		µL	*S. cerevisiae*	mL	*C. albicans*
AR 45 day	control	100	4.80 ± 0.26 ^a^	1.0	10.73 ± 0.71 ^a^
	NAA	^-^		1.25	10.63 ± 0.55 ^a^
	IBA	^-^		NT	
AR 60 day	control	250	4.63 ± 0.15 ^a^	1.25	5.83 ± 0.15 ^b^
	NAA	-		1.25	9.83 ± 0.29 ^a^
	IBA	-		1.0	10.00 ± 0.50 ^a^
ethanol, 20 µL		4.00 ± 0.10 ^b^		-	
fluconazole, 20 µL		9.00 ± 0.5 °C		5.00 ± 0.1 °C	
Saponin from *Quillaja bark*, 25 µL		14.77 ± 0.68 ^d^		16.17 ± 0.76 ^d^	

- No activity; NT—not tested. In vitro adventitious root (AR) culture, 45 or 60 day duration of cultivation, control ½ MS without auxin, IBA indole-3-butyric acid, NAA α-naphthalene acetic acid at a concentration of 1 mg/L. The data are presented as M ± SE. Significant differences in the same column are represented by different letters (a–d) *p* = 0.05.

## Data Availability

No new data were created or analyzed in this study. Data sharing is not applicable to this article.

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
