# Peer review of "Total Content of Saponins, Phenols and Flavonoids and Antioxidant and Antimicrobial Activity of In Vitro Culture of Allochrusa gypsophiloides (Regel) Schischk Compared to Wild Plants"

_plants, 2023, doi:10.3390/plants12203521_

Round 1

Reviewer 1 Report

Author did a work on "Total content of saponins, phenols and flavonoids, and antioxidant and antimicrobial activity of in vitro culture of Allochrusa  gypsophiloides (Regel) Schischk. compared to wild plants". Some of the minor corrections that are required

1. what is ICO50?

2. Too many keywords are used try to use 3-5 keywords only.

3.  Discussion is well written

4. Typo errors in Table 2 and 3 correct it.

5. what is 70о ethanol?

6. Make the conclusion in brief. 

2,

 Minor editing of the English language required

Author Response

Dear Reviewer!

Thank you very much for taking the time to review this manuscript. Please find the detailed responses in the re-submitted files

Reviewer 2 Report

See the attached file for my comments.

Need extensive English editing service

Author Response

Dear Reviewer!

Thank you very much for taking the time to review this manuscript. Please find the detailed responses and the corresponding revisions in the re-submitted files

Round 2

Reviewer 2 Report

Happy with the response. 

Author Response

Dear Reviewer! Thank you very much for taking the time to review this manuscript.